# The impact of the COVID-19 pandemic on the research activity and working experience of clinical academics, with a focus on gender and ethnicity: a qualitative study in the UK

Gabrielle M Finn [ID],[1] Paul Crampton [ID],[2] John AG Buchanan,[3] Abisola Olatokunbo Balogun [ID],[2] Paul Alexander Tiffin,[4] Jessica Elizabeth Morgan [ID],[5,6] Ellie Taylor,[2] Carmen Soto [ID],[7] Amelia Kehoe[2]

For numbered affiliations see end of article.

**Correspondence to**
Professor Gabrielle M Finn; gabrielle.finn@manchester.ac.uk

## ABSTRACT

**Objective** To investigate the impact of the COVID-19 pandemic on the research activity and working experience of clinical academics, with a focus on gender and ethnicity.

**Design** Qualitative study based on interviews and audio/written diary data.

**Setting** UK study within clinical academia.

**Participants** Purposive sample of 82 clinical academics working in medicine and dentistry across all career stages ranging from academic clinical fellows and doctoral candidates to professors.

**Methods** Qualitative semistructured interviews (n=68) and audio diary data (n=30; including 16 participants who were also interviewed) collected over an 8-month period (January–September 2020), thematically analysed.

**Results** 20 of 30 (66.6%) audio diary contributors and 40 of 68 (58.8%) interview participants were female. Of the participants who disclosed ethnicity, 5 of 29 (17.2%) audio diary contributors and 19/66 (28.8%) interview participants identified as Black, Asian or another minority (BAME). Four major themes were identified in relation to the initial impact of COVID-19 on clinical academics: opportunities, barriers, personal characteristics and social identity, and fears and uncertainty. COVID-19 presented opportunities for new avenues of research. Barriers included access to resources to conduct research and the increasing teaching demands. One of the most prominent subthemes within 'personal characteristics' was that of the perceived negative impact of the pandemic on the work of female clinical academics. This was attributed to inequalities experienced in relation to childcare provision and research capacity. Participants described differential experiences based upon their gender and ethnicity, noting intersectional identities.

**Conclusions** While there have been some positives afforded to clinical academics, particularly for new avenues of research, COVID-19 has negatively impacted workload, future career intentions and mental health. BAME academics were particularly fearful due to the differential impact on health. Our study elucidates the direct and systemic discrimination that creates barriers to women's career trajectories in clinical academia. A flexible,

## STRENGTHS AND LIMITATIONS OF THIS STUDY

⇒ The study used both audio and written diary entries to enable exploration of sensitive issues.
⇒ Telephone and online interviews enabled data collection to continue during the pandemic.
⇒ The research team was diverse in terms of gender, ethnicity and experience.
⇒ The study was limited to the UK.
⇒ The clinical academics were limited to doctors and dentists.

strategic response that supports clinical academics in resuming their training and research is required. Interventions are needed to mitigate the potential lasting impact on capacity from the pandemic, and the potential for the loss of women from this valuable workforce.

## INTRODUCTION

The workforce presence of clinical academics is vital to healthcare to ensure that scientific knowledge underpins advances in medical education, translational research and patient outcomes. Their low number and lack of senior pipeline progression is alarming, particularly so for under-represented groups including women and ethnic minorities.[1] The COVID-19 pandemic has further exacerbated the situation through rapid redeployment, burn-out, reinforcement of gender and racial stereotypes, and financial inequity.[2]

Over 1500 academic trainees in England were estimated to be redeployed to clinical work during the COVID-19 pandemic.[3] This equates to more than 90% of those on the Integrated Academic Training pathway.[3] Following the closure of high-risk aerosol-generating dental services, some junior hospital dentists and dental clinical

academics were also redeployed to provide support care for hospital services.[4] The potential impact on trainees' clinical experience has been recognised.[5] In terms of formally monitoring and permitting advancement in post-graduate training, 'no fault' Annual Review of Competence Progression outcomes (10.1 and 10.2) have been introduced where acquisition of competencies has been delayed owing to COVID-19.[6] Meanwhile, there has been a research paradox whereby some research institutions have closed, while others have adapted and embraced COVID-19-specific research.[7] Further issues highlight the consumption of academic resources that may prove ineffective or wasteful owing to haste, rapid redesign of online teaching materials, and inadequate infrastructure and design.[8] With the current prioritisation of COVID-19 studies, new clinical trials have been paused[9] with detrimental impacts on clinical academics working on non-COVID research.

Prior to the pandemic, it was recognised that women in medicine tended to take on more 'service work' and had less protected time for research than their male counterparts.[4] In addition to low representation of minority groups, the lack of role models provides a further barrier.[5] Collectively, gender and racial stereotypes, childcare and homeworking all put further pressure on an already stretched workforce, with women now publishing less.[6 7] Losing further women and minority groups from this fragile workforce could run the risk of long-term reduced diversity in medical education, less female-relevant clinical research and ultimately have a negative impact on patient outcomes, in addition to having less women visible in senior positions.[8 9]

We sought to explore discrimination based on protected characteristics in the clinical academic workforce, at various stages of the pipeline during the pandemic. We aimed to understand the barriers, enablers and interventions to facilitate clinical academic careers, thereby to develop key recommendations to support and reduce attrition of these exceptionally talented individuals. We were particularly interested in the experiences of women and clinical academics from different ethnic groups.

## METHODS
### Context
Prior to the COVID-19 pandemic, the authors were engaged in a multifunder research project exploring barriers and facilitators to a career in UK clinical academia.[10 11] As data were iteratively analysed, the impact of COVID-19 on clinical academia progressively shaped the experiences of the participants and relevant themes emerged. In order to assess the impact of COVID-19, the data collected after the first confirmed case within the UK (29 January 2020) until the end of September 2020 were isolated for analysis. This period included the first period of significant national restrictions ('lockdown') from 26 March to a phased reopening in June.

This study was conducted within a social constructivism orientation, additionally informed by the multiple principles of feminist theory. Feminist theory holds a constructivist ontology.

### Patient and public involvement
The study had a steering group which included patient and public representation, provided by Health Watch.

### Study design and recruitment
Qualitative interviews and longitudinal qualitative audio and written diary entries were used. Purposive sampling was undertaken, using social media, funder mailing lists and snowballing. Inclusion criteria specified participants were a doctor or dentist, based within the UK and working as or towards becoming a clinical academic. For the purposes of the study, a 'clinical academic' was defined by individuals themselves and how they perceived their roles. Recruitment was through various avenues; a website, emails sent on behalf of the research team by stakeholder organisations; social media recruitment; and 'snowballing'. Authors had no relationship with participants prior to the study.

### Data collection
Data were collected using (1) Audio and written diary entries, and (2) Semistructured interviews. Diaries enabled participants to discuss their views on a wide range of factors related to clinical academic careers, and latterly their experiences of working during COVID-19. The diary method enabled researchers to collect 'novel' and timely data during the transition to lockdown phase, a critical moment in healthcare. Many diaries were recorded immediately after events, such as discriminatory remarks being made in a workplace. Participants were invited to submit audio diary entries (MP3 files) via email or attached to an encrypted message. Submissions were sent to a professional account linked to the project following best practice.[12] In order to promote engagement, participants were also permitted to write their diary entry.[12] Participants were able to submit diaries as frequently as they wished, each acknowledged by the researchers to maintain rapport and engagement. Participants were issued with a guide to follow which was informed by a concurrent systematic review.[9] The guide suggested frequencies of entries, instructions for submission and indicative content (online supplemental appendix 1). Reminder emails were sent monthly. Interviews were conducted via videoconferencing or telephone with the use of a topic guide. Audio diary files and interviews were recorded and transcribed verbatim. All participants were given an electronic information sheet and consent form prior to the recording. Participants returned written consent via the electronic consent form; this was also confirmed verbally at the start of each interview. Interviews were conducted by GF, PC, AK, JAG and ABK. Authors are educationalists (GF, PC, AK, JB, PT), clinicians (ET, JM, PT, JB, CS) and qualitative researchers (all). The researchers hold a range

of qualifications including PhD, MD, MBChB, MMEd, PGDipMEd, MRCpsych, FRCP, FAcadMEd, FRCPsych and MSc. The research team consisted of clinicians and non-clinicians, clinical academics at varying stages, expert qualitative researchers to novices, women and men, and a mix of ethnicities.

## Data analysis

Data were thematically analysed[13 14] using reflexive thematic analysis within NVivo (V.10. QSR International; 2012). Coding was undertaken by GF, PC, AK, JAG, ABK and ET. Thematic analysis was chosen because it is an appropriate method for seeking to understand experiences, thoughts or behaviours across a data set.[15] The authors read the data in its entirety and developed a codebook as a tool to assist in the analysis of the large data set.[16] The six-step process of thematic analysis was followed: (1) Data familiarisation, (2) Generating initial codes, (3) Searching for themes, (4) Reviewing themes, (5) Defining and naming themes, and (6) Producing the report.[14 17] Both inductive and deductive approaches were taken, with deductive analysis based on existing theory including maternal wall bias, feminist theory and intersectionality.[18–20] A full coding tree can be accessed in a funder report.[10] Authors engaged in a process of negotiation to refine codes and themes, before using member checking with a subset of participants. Authors were reflexive, recording reflexive journals and acknowledging their biases and presuppositions. Team meetings were regularly held to discuss reflexivity. Since the study data were confined to a limited collection period during the pandemic, it was felt that this would not be of sufficient duration for meaningful temporal analysis. Instead, researchers interpreted the data as being snapshots and in-the-moment reflections relating to the impact of the pandemic. In order to bring participants' experiences over time, cross-sectional themes together and demonstrate intersectionality, case studies from two participants are provided (box 1). Authors collected data until such time that they felt meaningful analysis and interpretation had been completed—in line with contemporary literature[21–23]; we do not subscribe to the notion of saturation but rather took a pragmatic approach in collecting data to answer the question, within the constraint of time and resource. Further, we noted that information power.[23]

## RESULTS

### Demographics

Of 82 individual participants, 14 provided audio diary data only, 52 were interviewed only, and 16 were both interviewed and provided audio diary data. Table 1 provides an overview of the descriptive statistics of the participant sample and the data collected. The research team received 134 diary entries, with a mean of 5 entries per participant. No participant withdrew or refused to participate.

---

**Box 1    Case examples of the initial impact of COVID-19 on participants with intersecting identities (pseudonyms used)**

**Case study 1: Asian, Mother, Muslim, Medical Registrar/ Doctoral Fellow**

Zainab, a Medical Registrar in Neurology and Doctoral Fellow is a married, Asian, Muslim, mother of three young children. She is 'proudly intersectional'. Zainab works full time and spends about 90% of her time on her academic work. She has recently put her details down to be called to the front lines and is currently on the stand-by list. As an Asian doctor, living and working in a city, she feels particularly vulnerable and has fears about her risks of contracting COVID-19. Her anxieties regarding the virus increased over time as research linked black, Asian, and minority ethnic (BAME) ethnicity to outcomes. She knows of a doctor who died on the front line, they were a relative of one of her closest friends. Although she realises that susceptibility to the disease is multifactorial, she can see that people from minority ethnic groups are disproportionately affected. According to her, *'it's not known why(there's a disproportionately higher number)I think it is multifactorial whether there's underlying kind of health vulnerabilities that are more prevalent in certain ethnic groups that make them more vulnerable or whether there's more complex issues to that'.* Despite the negative implications that the COVID-19 pandemic and lockdown have brought, she finds a great deal of positives from the supportive academic institution.

**Case study 2: White, Mother, Clinical Lecturer**

Rose, a Medical Registrar and Academic Clinical Lecturer, has been working at home where she is married to a clinical professor with two children. Rose works part-time. She is struggling with the balance of family and work commitments, stating that lockdown has reinforced sexist gender roles. Rose feels that her research has taken a back-seat while her husband's work has been prioritised, *'I genuinely think he thinks his work is more important, I genuinely think that's what he believes'.* She reports a decline in mental health due to the pressures of lockdown and family dynamics, *'I think that COVID-19 has been a disaster for feminism…the disagreements that I've had with (husband) have been over feeling like a 1950s housewife suddenly, then to suddenly have to provide home schooling children and trying to stay sane, stay safe, keep everybody okay, whereas he still is doing exactly what he does (work)…so I think that, that, that's my big worry is that actually COVID-19's been a step back'.* Rose's anxiety about her research profile increased the longer the lockdown restrictions were in place.

---

## Themes

The analysis revealed four major themes, each with subthemes (table 2), relating to the initial impact of COVID-19 on clinical academic careers; (1) Opportunities, (2) Barriers, (3) Personal characteristics and social identity, and (4) Fears and uncertainty. Intersectionality and the differential impact of ethnicity and gender on the experiences were noted across all themes.

Due to the extensive data set, only selected themes are presented in continuous prose; these have been chosen due to their pertinence to the research. The analysis found the COVID-19 experiences of medical and dental clinical academics relatively similar, differences were noted when protected characteristics such as gender and ethnicity were considered.

**Table 1** Demographic breakdown of participants

| | Audio diaries | | Interviews | |
|---|---|---|---|---|
| | Total (n=30) | % | Total (n=68) | % |
| **Profession** | | | | |
| Medicine | 24 | 80.0 | 57 | 83.8 |
| Dentistry | 6 | 20.0 | 11 | 16.2 |
| Mean age, years | 39 | | 40 | |
| Age range, years | 27–74 | | 28–51 | |
| **Gender** | | | | |
| Male | 10 | 33.3 | 28 | 41.2 |
| Female | 20 | 66.6 | 40 | 58.8 |
| **Predominant clinical work area** | | | | |
| Primary | 7 | 23.3 | 19 | 27.9 |
| Secondary | 11 | 36.7 | 30 | 44.1 |
| Tertiary | 12 | 40.0 | 17 | 25.0 |
| Did not disclose | 0 | 0 | 1 | 1.5 |
| **Employment status (overall)** | | | | |
| Full time | 24 | 80.0 | 65 | 95.6 |
| <Full time | 6 | 20.0 | 3 | 4.4 |
| Did not disclose | 0 | 0 | 0 | 0.0 |
| **% of hours spent on academic work** | | | | |
| 100% | 5 | 16.7 | 8 | 11.8 |
| 50% | 19 | 63.3 | 3 | 4.4 |
| <50% | 6 | 20.0 | 57 | 83.8 |
| Did not disclose | 0 | 0 | 0 | 0.0 |
| **Out of programme for research** | | | | |
| No | 19 | 63.3 | 55 | 80.9 |
| Yes | 5 | 16.7 | 3 | 4.4 |
| Not applicable | 6 | 20.0 | 10 | 14.7 |
| **Ethnicity** | | | | |
| Asian | 2 | 6.7 | 11 | 16.2 |
| Indian | 2 | 6.7 | 3 | 4.4 |
| Middle Eastern | 1 | 3.3 | 2 | 2.9 |
| Black | 0 | 0 | 3 | 4.4 |
| White Caucasian | 24 | 80.0 | 47 | 69.1 |
| Did not disclose | 1 | 3.3 | 2 | 2.9 |
| **Marital status** | | | | |
| Divorced | 2 | 7.0 | 0 | 0.0 |
| Long-term relationship (not married) | 2 | 7.0 | 7 | 10.3 |
| Married | 23 | 77.0 | 56 | 82.4 |
| Single | 3 | 10.0 | 4 | 5.9 |
| Did not disclose | 0 | 0 | 1 | 1.5 |
| **Sexuality** | | | | |
| LGBTQIA+ | 1 | 3.3 | 4 | 5.9 |
| Heterosexual | 25 | 83.3 | 62 | 91.2 |

Continued

**Table 1** Continued

| | Audio diaries | | Interviews | |
|---|---|---|---|---|
| | Total (n=30) | % | Total (n=68) | % |
| Did not disclose | 4 | 13.3 | 2 | 2.9 |
| **Disability** | | | | |
| No | 28 | 93.3 | 67 | 98.5 |
| Yes | 2 | 6.7 | 1 | 1.5 |
| **Number of children/dependents** | | | | |
| 0 | 7 | 23.3 | 27 | 39.7 |
| 1 | 6 | 20.0 | 11 | 16.2 |
| 2 | 11 | 36.7 | 25 | 36.8 |
| 3 | 4 | 13.3 | 2 | 2.9 |
| 4 | 2 | 6.7 | 1 | 1.5 |
| Did not disclose | 0 | 0 | 2 | 2.9 |
| **Pregnant** | | | | |
| Yes | 0 | 0 | 1 | 1.5 |
| Did not disclose | 1 | 3.3 | 7 | 10.3 |
| No | 29 | 96.7 | 60 | 88.2 |
| **Current clinical academic career level** | | | | |
| Doctoral Fellow/PhD student | 10 | 33.3 | 13 | 19.1 |
| Academic Clinical Fellow | 6 | 20.0 | 14 | 20.6 |
| Academic Clinical Lecturer | 6 | 20.0 | 14 | 20.6 |
| Senior Clinical Lecturer and above (including Deans and Programme Directors) | 6 | 20.0 | 27 | 39.7 |
| Did not disclose | 2 | 6.7 | 0 | 0.0 |
| **Current grade within clinical role** | | | | |
| Clinical Fellow | 3 | 10.0 | 1 | 1.5 |
| Registrar (Medical/Dental) | 16 | 53.3 | 22 | 32.4 |
| General Practitioner (Medical/Dental) | 4 | 13.3 | 6 | 8.8 |
| Medical/Dental consultant | 6 | 20.0 | 30 | 44.1 |
| Medical researcher | 1 | 3.3 | 0 | 0.0 |
| Did not disclose | 0 | 0 | 1 | 1.5 |
| **Location** | | | | |
| East of England | 1 | 3.3 | 2 | 2.9 |
| Midlands | 4 | 13.3 | 11 | 16.2 |
| North-East England and Yorkshire | 11 | 36.7 | 26 | 38.2 |
| North-West of England | 2 | 6.7 | 4 | 5.9 |
| South-East of England | 6 | 20 | 16 | 23.5 |

Continued

**Table 1** Continued

| | Audio diaries | | Interviews | |
|---|---|---|---|---|
| | Total (n=30) | % | Total (n=68) | % |
| South England | 4 | 13.3 | 2 | 2.9 |
| Wales | 2 | 6.7 | 1 | 1.5 |
| Scotland | 0 | 0 | 5 | 7.4 |
| Did not disclose | 0 | 0 | 1 | 1.5 |
| Place primary health qualification awarded | | | | |
| UK | 30 | 100 | 64 | 94.1 |
| IMG | 0 | 0 | 2 | 2.9 |
| EEU | 0 | 0 | 1 | 1.5 |
| Did not disclose | 0 | 0 | 1 | 1.5 |
| Total number of diary entries | 134 | | N/A | |
| Number of written entries | 26 | | N/A | |

EEU, European Economic Union; IMG, International Medical Graduate.

## Theme 1: Opportunities and enablers

The pandemic presented many opportunities and enablers. These spanned academic and clinical work as well as participants' personal lives. Subthemes broadly fell into three categories; (1) New opportunities for research, (2) Support, (3) Maintenance of normality and (4) Technological advances supporting remote working.

### New emerging opportunities

Participants recognised the immediate opportunity to explore new avenues of research related to the pandemic.

> So I've been asked to, and volunteered to, be part of the Research Ethics
>
> Committee Coronavirus Response which means that every week now I've had twenty-four hour turnaround for a coronavirus study of some sort or another … I have been involved with developing two national and one international *coronavirus studies over this time… an enormously exciting but terrifying roll out of research…* [Male Medic - diary]
>
> …there might be a positive if you were, you know, an Oxford University clinician looking at vaccine so you'd obviously you get a lot of funding and, you know, good opportunity to, to research COVID-19 based things… I suppose if you've never done a clinical trial before and you were a clinical academic, getting involved in some of the trials that they're doing at the moment, like the Recovery Trial, that could be a positive for you if you're in that particular situation… [Female Medic – interview]

Associated with this was the need for rapid research dissemination, providing clinical academics with a chance to develop their portfolio. This resulted in the forging of new research collaborations supported by the quick turnaround of ethical approval in order to roll out impactful research. More senior clinical academics reported opportunities for career development such as being invited to serve on research ethics committees. In addition to the new research territory, COVID-19 presented a new lens through which participants were able to think about their existing research.

For some, salient among the opportunities was the free time which was created during the pandemic, particularly due to the absence of social commitments and commuting. Some clinical academics found this spare time to be an opportunity to focus on academic research activities and the production of impactful research. Others felt this was as an opportunity to focus on clinical work, as well as family and household commitments.

> I guess it will give me a chance to focus on, you know, having a bit more time at home with family and, and just doing clinical work, which I think, you know, is what I wanted to focus on for the moment. [Female Medic – Audio Diary]

### Support

Participants presented examples of the positive impact of support networks locally, nationally and internationally. The abrupt emergence of COVID-19 and subsequent lockdown meant that many clinical academics halted their academic work and returned to clinical practice. This created an element of uncertainty, and in some cases anxiety about completing their academic work. However, a widely circulated statement from the UK's main funding bodies reassuring clinical academics that they would support the extension of research post-COVID was well received.

> The (funder) have said that they will support extensions if they're needed so I do have that to kind of fall back on. [Female Medic – Audio Diary]

An immense amount of support was provided by academic organisations at the beginning of the pandemic. Support from supervisors, departments and funding bodies was identified as enabling and empowering for participants. For example, the provision of childcare nurseries, enabling clinical academics to return to full-time clinical practice when required.

### Maintenance of normality

'Maintaining the status quo' during the pandemic was an enabler for many. From the beginning of the pandemic many professions have seen large proportions of redundancy. However, this has not been the case for many of the clinical academics that participated in this study. Clinical academics who have had the flexibility of undertaking either clinical or academic work during the pandemic, were grateful and felt generally lucky to have a job during COVID-19 and one they could go back to

**Table 2** Themes and subthemes from audio diary entries

| Opportunities | Barriers | Personal characteristics and social identity | Fears and uncertainty |
|---|---|---|---|
| ⇐ ⇐ ⇐ **Intersectionality** ⇒ ⇒ ⇒ The impact of gender, ethnicity and associated intersectional identities were pertinent across all themes | | | |
| **New emerging opportunities**<br>► New research avenues<br>► New collaborations<br>► COVID offers a new lens through which to think about existing research<br>► Increased capacity and opportunity to write | **Lockdown**<br>► Working at home with children<br>► Work-life balance<br>► Parenting responsibilities<br>► Positive discrimination<br>► Gender roles<br>► Impact on physical and mental health from working at home<br>► Difficulty maintaining momentum<br>► Decline in academic outputs<br>► Isolation<br>► International work difficult<br>► Physical barriers impacting on work (eg, PPE, remote consultations)<br>► Demands from teaching responsibilities<br>► Loss of resource | **Being a clinical academic**<br>► Increased perceived importance of clinical academics<br>► Sitting in neither camp | **Returning to the front line**<br>► Responding to the call to arms<br>► Contracting COVID-19<br>► BAME ethnicities increased susceptibility<br>► Anxiety<br>► Appropriateness of skill set for return to clinical duty<br>► Inability in the short term to conduct research<br>► Impact on long-term research career |
| **Support**<br>► Funder support (extensions and network)<br>► Supportive networks (local, national, international)<br>► Provision of childcare for keyworkers<br>► Camaraderie<br>. | | **Protected characteristics**<br>► BAME<br>► Maternal status<br>► Gender stereotypes<br>► Intersectional identities | **Misconceptions and absence information**<br>► Funding<br>► Extensions<br>► Lockdown restrictions being lifted<br>► Fiscal impact and predicted recession<br>► Portfolio development due to loss of research capacity<br>► Future impact of COVID<br>► Uncertain job market |
| **Maintenance of normality**<br>► Job security<br>► Maintenance of structure<br>► Workload maintained | **Demand from teaching responsibilities**<br>► Switch to online teaching<br>► Lack of support for teaching activity<br>► Teaching is devalued | | |
| **Technological advances supporting remote working**<br>► Maintenance of networks<br>► Inclusive approach to working | **Loss of resource**<br>► Loss of outputs<br>► Fiscal barriers<br>► Loss of research team and associated HR issues<br>► Decrease in academic capacity due to clinical workload<br>► Lack of research supervision<br>► Practical barriers (eg, data storage)<br>► Absence of pastoral support | | |

BAME, black, Asian, and minority ethnic; HR, Human Resources; PPE, Personal Protective Equipment .

post-COVID. Opportunities that strengthened the resolve of participants included the option to flex hours between roles. Specifically, it was also noted that going back to clinical practice full-time was perceived as advantageous due to more structure compared with academic work.

Additionally, the retention of social interaction, structure and workload was seen as an opportunity.

In these times of COVID-19 some people have kind of lost their structure and workload, so I feel grateful

that hasn't happened to me. [Male Medic – Audio Diary]

## Technological advances supporting remote working

Participants advocated the use of videoconferencing platforms, citing them as enablers to maintain networks and collaborations during the pandemic as well as the inclusivity afforded by digital spaces.

…my colleagues have become much more able to work online using Microsoft Teams and there have been more meetings recorded and more webinars which meant that as a part-time person I've been able to attend some of the things that previously I used to enquire and they used to say that this is not recorded and it is not being broadcast, so that has been excellent. [Female Medic – Interview]

### Theme 2: Barriers

There were multifaceted barriers related to the direct consequences of the immediate impact, reactive and mid-term barriers, and longer-term implications. Three subthemes were identified; (1) Lockdown, (2) Demand of teaching responsibilities and, (3) Loss of resource.

### Lockdown

Most of the barriers described by participants related to the negative consequences of lockdown. A barrier evidenced by particularly emotive data was that of working at home. This entailed the need to balance the multiple commitments related to childcare, housekeeping and productivity. There were far-reaching consequences of the ways in which working from home impacted on participants' roles. Childcare was not seen as compatible with a clinical academic role. Both clinical (eg, telephone clinics and clinical administration) and academic work was often taking place at home, leading to feelings of isolation.

I've spent a lot of time doing very intensive blocks of clinical work and very intensive blocks of research, research policy into practice work…I realised it's a Thursday today, I realised I had worked for 3 weeks solid and I'd not had a full day off in that time and most of those days were ten or twelve hour days…and I have now worked a further eleven days, with the last 4 days being fourteen hour days, without a break. [Male Medic- Interview]

Mental and physical health outcomes were described as a consequence of COVID-19, particularly due to the increased workload.

Generally, the COVID-19 experience has been exhausting on many levels; emotionally from the worry about the number of people who will die, worry about being redeployed, worry for my children's well-being. [Male Medic- Interview]

My anxiety is elevated due to being Asian, will I get sick and lose time for research? [Female Medic-Interview]

Concerns arose over how to effectively manage competing time demands. Some also spoke about a loss of 'downtime' and feeling 'stuck in limbo'. The shift in work-time meant that participants had less time for recreational activities which impacted on well-being.

Both men and women were impacted by the move to homeworking and balancing childcare with full-time work. The time pressures were apparent in many situations where participants struggled to navigate competing demands.

… the Pro Vice Chancellor had sent an email saying how exciting it was that there's a female President at the Medical School, a female Dean, a female Pro Vice Chancellor of Medicine and a female Vice Chancellor at University, and that made me reflect what would happen if I sent a similar email delighting the fact that there's males in all of those roles. I imagine there'd be some disdain… they're powerful and take revenge…I don't think that's appropriate. [Male Medic – Audio Diary]

There were also concerns over positive discrimination in how gender roles were recognised, with a particular focus on senior positions. Such comments alluded to notions of male fragility and toxic masculinity, although the intent cannot be stated in absolute terms.

…I do sometimes think that even, males might be at a, particularly at the end of, top end of their career, might be at more, slight disadvantage because, particularly with Athena SWAN there's obviously an emphasis, and maybe rightly so, in supporting women, you know, particularly take senior posts like chairs and therefore I do think sometimes it's almost got the point where it's the first discrimination where men might be overlooked, you know, because there's a keenness to promote women, and I, in a sense I understand the rationale for that, and it might get worse with the pandemic. [Male medic- interview].

it's kind of bordering on positive discrimination around research professorships, I think men are needed… [Male medic- interview].

### Demands from teaching responsibilities

The analysis identified the negative impact of having to rapidly produce teaching and assessment materials for online delivery, frequently coupled with a lack of support and appreciation, as well as the time consumed. In many cases, this was in addition to clinical responsibilities.

As clinical academics we are scavenging time in between the clinical parts to crack on and do our academic work. [Male Medic – Audio Diary]

There was worry over a future lack of recognition within portfolios for teaching activities undertaken, and a frustration over negotiating the various new technologies required to deliver teaching.

### Loss of resource

The impact of COVID-19 has been significant for participants and extends from fiscal and time to human resource issues. There were concerns over the loss of time in relation to academic outputs and not fulfilling previously arranged activities. A lack of overall supervision was noted, including much needed pastoral support.

> There is an absence of pastoral support. Supervision is different, it's about skills and development not always about just how you feel. Sometimes you need a shoulder to cry on. [Female Medic – Audio Diary]

Given the measures in the academic field where outputs are often used as a productivity marker, the participants were unsure how they could compensate loss of time and the ways in which their activities could be demonstrated and valued.

Tangible barriers were highlighted in the monetary implications for how loss of time could be compensated. Participants made calls for funders to extend time and funding for projects that otherwise may have been completed on time.

> It's bloody terrifying when it comes to applying for new stuff though because the deadlines haven't really moved much and there's no head space and there's no time to get the stuff really done properly and that's really scary but we've got to get the clinical work done first. It's the only way. [Male Medic – Audio Diary]

Many universities initiated a recruitment freeze which meant in some cases that methodological expertise was lost. As clinical academics are often on a set educational employment path for a discrete time period this created disruption to long-term career plans.

### Theme 3: Personal characteristics and social identity

A key theme revealed was the impact of COVID-19 on participants' personal characteristics and social identity. This theme specifically considers how individuals perceived themselves as being clinical academics and the issues associated with the label 'clinical academic'.

### Ambivalence and identity of a clinical academic

Some participants were conflicted about their identity as clinical academics and often experienced 'imposter syndrome'. Imposter syndrome, as described by the participants, refers to a feeling that they were not adequately qualified to be called clinical academics. Others expressed pride in being classed as clinical academics, however, they expressed that their identity as clinical academics was misunderstood by clinical and academic colleagues. Participants described problems experienced with negotiating both clinical and academic identities during the pandemic.

> Trying to tussle where you fit between university and clinical world at the moment is, is also hard. [Male Medic– Audio Diary]

Participants highlighted that in their working life they do not sit neatly in either category, which makes it difficult to identify as either an academic and/or a clinician. This feeling has been intensified by the pandemic. Most professionals have an educational background where they have developed peer support and networks allowing for guidance, but for clinical academics, their varied pathways means they often lack this important support network.

### Protected characteristics

Protected characteristics of clinical academics, particularly the black, Asian, and minority ethnic (BAME) community who have been more widely affected by COVID-19, was a key subtheme identified. Increased anxiety was highlighted and participants who were from BAME backgrounds expressed fear about returning to clinical work. Muslim clinical academics also expressed the difficulty they faced due to Ramadan taking place during the already difficult time. The focus was largely on physical well-being due to heavy workload alongside being unable to eat or drink.

Gendered differences were evident throughout this study, with female participants specifically expressing the struggles of maternal identity, highlighting problems faced with childcare responsibilities as well as having to share these with their partners alongside work commitments.

> I'm not formally planning to carry on with academia and that's a decision that's kind of come about because of, you know, childcare responsibilities and I just felt that I was going to be stretched too thin if I was trying to be a mum and be a GP and be an academic as well. [Female Medic– Audio Diary]

Female participants felt that due to gender stereotypes from partners and work colleagues they were having to take a leading responsibility in childcare during this time.

> I think that COVID-19 has been a disaster for feminism…the disagreements that I've had with (husband) have been over feeling like a 1950s housewife suddenly then to suddenly home schooling children and trying to stay sane, stay safe, keep everybody okay, whereas he still is doing exactly what he does, work… [Female Medic– Audio Diary]

While participants noted intersectionality, their narratives were mostly differentiating their experiences due to gender. Women explicitly stated that they became less tolerant of the gender issues raised as time in lockdown progressed.

Women reported the emotional turmoil associated with being primary caregivers.

> Yet another day of working from home while helping with young kids who are upset at the disruption to their lives. The most difficult thing is trying to

calm my children's' worry over COVID-19. [Female Dentist– Audio Diary]

### Increased perceived importance of clinical academics

Despite the tensions that have been caused by the pandemic, some noted that the work and identity of clinical academics have actually been strengthened. COVID-19 has required rapid research to be undertaken and clinical academics are perfectly suited to this opportunity. The fact that qualified clinical academics have a foot in both research and clinical practice has facilitated a renewed appreciation of their identity during this pandemic.

### Theme 4: Future fears and uncertainty

This theme related to numerous unanswered questions of participants including their fears and anxieties. Returning to the clinical environment was one major cause of anxiety, as well as the future of their research and funding.

### Returning to the front line

Discourse regarding a full return to clinical practice used many analogies to warfare. One source of internal conflict for participants was their inability to respond positively to the 'call to arms', particularly where they felt they had deskilled in some areas due to their focus on academia in recent years. Participants described requests to abandon research and assume clinical roles. Added to this anxiety was an inability to manage workload and family life.

> Well all of the academic trainees have been pulled on to the full-time clinical work. As clinical academics we are scavenging time in between the clinical parts to crack on and do our academic work and certainly emotionally I'm finding it enormous strain, mainly because the clinical work itself has not so much increased in the intensity but the massive decrease in staff numbers, partly because of illness but a lot because of isolation and shielding requirements that have gone on, has hugely increased the amount of work that all of us that are at work are doing. [Male Medic– Audio Diary]

Additionally, the fear of contracting COVID-19 was expressed, with the majority of participants having to return to full clinical practice. Clinical academics who expressed fear of contracting COVID-19 were those with predisposing factors and BAME backgrounds. Additionally, knowing someone who had died on the front lines increased clinical academics' perception of their own risk.

### Misconceptions and concerns about the future

From the onset of the pandemic, participants have reported concerns over their progression and future careers. Participants also aired concerns about the lockdown restrictions being lifted prematurely, stating that this could have further negative implications on their clinical and academic work. Uncertainty around COVID-19 has caused misconceptions, concerns and scaremongering, ranging from the prospect of more lockdowns to the potential for a return to full-time clinical work only. There has also been an absence of concrete information for clinical academics. Among participants, there is a general fear and uncertainty about the fiscal impact on research funding and the future clinical academia.

> Unfortunately my grant holders haven't sort of committed themselves to being able to extend my funding due to various reasons, so yeah, so there's uncertainty with that as well. [Female Medic– Audio Diary]

Issues such as pay cuts and a loss of staffing resources within academic departments caused further anxieties.

> The news on pay cuts is a bitter pill to swallow when working in a department where teaching commitments have increased, I have to say. Clinicians may decide to switch to clinical only in some cases. [Male Medic– Audio Diary]

These anxieties have been further intensified following a lack of professional and social networking opportunities, especially as the pandemic intensified and lockdowns progressed.

### DISCUSSION

This study presents a unique, qualitative data-driven reflection of the national clinical academic landscape during 9 months of the COVID-19 pandemic. The in-depth interview data are supported by real-time audio diaries. This multifaceted approach has facilitated access to 'periods of change and flux' particularly during the pandemic, and allowed the research team to capture the impact of COVID-19 in real time which facilitated accurate data capture.[24 25] The audio diaries are also advantageous owing to their ease of completion, which typically results in lower levels of attrition.[25] Participants were also permitted to submit written entries, but this was not deemed to be a limitation, especially as discourse analysis was not planned. Permitting flexibility also prevents diary entries from becoming formulaic.[25] While the authors recognise that there is often a self-selection bias associated with studies of this nature, participants were felt to be balanced in their presentation of barriers and enablers.[26] Audio diaries varied in length and thus depth, however, there was no variability in the richness of data. This study was funded and thus the qualitative nature stipulated—later we discuss ideas for future research using quantitative and survey approaches. Such work could add to our findings.

Clinical academics were impacted by COVID-19 in multiple ways leading to a wide range of barriers, challenges and opportunities underpinned by their personal and professional identities. The selflessness and dedication of the study participants to contribute to the clinical response to the COVID-19 pandemic is evident in the data. Clinical academics returning to the National

Health Service during the pandemic, had added value with respect to their research experience.[27]

Our findings identified the numerous perceived barriers to continuing academic activity within the family, academic and clinical contexts. What is clear is that pre-existing barriers to academic activity have become magnified during the COVID-19 pandemic. Although such barriers are not insurmountable, they have proved and continue to prove stressful for the participants. The impact on their future career plans is already evident. The restrictions on face-to-face contact, international travel, uncertainties over clinical and academic training and funding extensions, homeworking, and, in many cases, redeployment to front-line clinical duties have all impacted on academic activities. Tension was evident when academic trainees who switched to full-time clinical work and sacrificed their academic time described how they felt disadvantaged in comparison to trainees who have been able to maintain research.

Female participants described barriers that related to their gender, as well as for many, their maternal status, with such biases having been previously well documented in the literature.[18 20 28–30] Our findings in the themes (2 and 3) and case studies, suggest that the pandemic and resultant lockdown has, in many cases, exacerbated the discrimination faced by many women in clinical medical and dental academia. A survey of dental clinical academics in 2018, found that overall female representation in the academic grades in Dentistry is increasing—mirroring gender representation in dentistry as a whole.[31] Given our data, it is possible that the COVID-19 pandemic could have an even more significant negative impact on female Dental Clinical Academics going forward.

Theme 3 also identified a number of women in our sample who described feeling and being compelled to assume, to a greater degree than before the pandemic, gender stereotypical roles in the home. Specifically, when faced with competing responsibilities women appeared to feel under more pressure than their male partners to allocate more time to childcare and household tasks. Indeed, there was a recurrent motif of narratives relating to the '1950s housewife' stereotype within the family context.[32] In this regard, our findings are consistent with media accounts that the lockdown has placed a disproportionate burden on women, who frequently report taking on the bulk of the childcare, domestic duties and home-based education of school-aged children.[33 34] It has been postulated that many women are currently doing 'second and third shifts' with regard to housework and welfare after their first shift of childcare or paid work.[32] Prior to COVID-19, there was evidence that parental working determined that, on average, women spent two to three times as much time on care and housework compared with men.[35] Our findings suggest that these effects have become more pronounced as a result of the pandemic and are impacting significantly on female clinical academics. Our data may go some way towards explaining reported declines in publications by female academics.[36]

The findings raise an alarm that prolonged periods of lockdown could reverse general trends towards a more equitable distribution of household labour.[35] More concerningly, it may substantially put back efforts to achieve a more gender representative body of clinical academics, characterised by gross under-representation of women, explored by a recent systematic review.[9] This has been shown to not merely be a matter of social equality; the male dominance of science in general has served women's health poorly.[37]

Of note within our data was the indication from male participants that they were experiencing positive discrimination. We would not wish to overinterpret; however, our analysis of the participants' narratives alluded to some potential degree of toxic masculinity[38] and fragility. Women's continued fight for equality within the workplace can often lead to discomfort from those who have typically held most privilege within the discussion. Although the discomfort articulated from men in our sample was a minority, it was expressed with conviction, especially in terms of positive discrimination.

There are a number of steps that could be taken to mitigate the impact of the COVID-19 pandemic on clinical academia as a whole and the risk of reduced long-term clinical research capacity, especially the loss of female academics. Funders and employers could intentionally target support at those with caring responsibilities during lockdown and the following recovery period. As the pressing clinical need recedes, prioritising the return of female academics to their research activities may also offset the bias more generally experienced by female scientists both before and during the pandemic. While virtual networks of collaborators have organically sprung up in places during the lockdown, more of these could be strategically formed to counter the isolation felt by many academics. This support should also extend to academics who are primarily clinical educators, and have historically felt marginalised and undervalued.[39] Indeed, some of the participants in the present study voiced perceptions of a lack of support and appreciation from both their academic and clinical employers, and colleagues. Both existing and trainee clinical academics reported a perception that neither non-academic clinical colleagues or non-clinical academics fully appreciated the competencies and the time demands of their academic roles. This perception has been reinforced by the COVID-19 pandemic. Going forward, enhanced collaboration and empathy between University and Clinical staff and employers could be targeted to remedy this.

As the UK navigates its immediate response to COVID-19, it is important to consider the potential implications for those whose careers are most vulnerable and require additional nurturing. There needs to be increased flexibility in career pathways and expectations, along with a review of funding and support offered. Participants requested flexibility with existing funding to make changes to study protocols and costings to enable them to pursue opportunistic COVID-19-related research.

Leaders in academic institutions and funding bodies must be cognisant of the varying degrees to which clinical academics will have been able to mitigate for COVID-19 in their assessments of applications for funding and positions and must revise their expectations for dissemination and impact. Causal explanations of the issues faced during COVID-19 will be multifaceted and sensitive to personal context. Teaching commitments must be valued and accounted for, as well as physical and mental health. It has been shown that clinical academics create national wealth as well as health.[40] We are heading into an inevitable global financial recession, if not depression, likely to also be characterised by an increase in mental and physical health needs which will undoubtedly impact on the clinical academic workforce and funding.

The infrastructure for medical and dental clinical academia is well established, yet our data suggest that better support is required to develop future clinical academic roles and enhance the provision of evidence-based patient care in the post-COVID-19 era. Participants report substantiated concerns over access to funding and positions once the status quo is resumed.

Whatever interventions and strategies are implemented, heed needs to be taken to the findings of Laver *et al*, who reported that interventions within clinical academia that require those they are designed to support, so called 'bottom-up' approaches, to bear the brunt of the workload are less likely to succeed.[41] Their review advocates the use of 'top-down' approaches, led by change in practice at a management level. Finally, their stance was that 'something is better than nothing'.

Our findings offer further insight into the debate surrounding access to, and maintenance of, clinical academic careers.[42] We add weight to the evidence surrounding gender disparity. The well-documented decline in the clinical academic workforce, the subject of contemporary research,[9 43] could continue if strategies to support clinical academics are not implemented in the aftermath of the COVID-19 pandemic. Ongoing commitment to supporting and developing clinical academics is required, acknowledging the implicit biases facing women and parents in reaching their potential, and those with other competing demands on their time. Of paramount importance is the need for greater recognition of the value and contribution made by clinical academics to both the clinical and academic workforces to which they contribute. The reports from a recent national study by Finn *et al* presented a number of proposed interventions to support clinical academics in the immediate wake of the pandemic, in addition to recommendations for ensuring the long-term sustainability of the clinical academic pipeline. Headlines from qualitative data included the need for mentorship, flexibility and transparent processes, as well as recognition of the differential experiences of marginalised groups.[10]

Our findings also emphasise the need for guidance for aspiring clinical academics. As one remarked, 'I just need guidance on next steps. It is just so challenging to think about who might help me'. Furthermore, without urgent action COVID-19 has the potential to increase the gender divide seen within clinical academia, and as demonstrated within recent studies.[40–42] It is hoped that the narratives in this study initiate a dialogue between clinical academics, funders and institutions to successfully navigate the clinical academic pathways moving forward. Support and mentoring may be key to ensuring the clinical academic pipeline.[44]

## Future research

Future research could include collecting quantitative data to track the career progression, publication rates and grant successes for clinical academics to look for patterns, trends and potential biases based on their protected characteristics and availability of support or mentoring. Survey data could supplement the findings of this study.

**Author affiliations**
[1]Division of Medical Education, School of Medical Sciences, Faculty of Biology, Medicine and Health, The University of Manchester, Manchester, UK
[2]Health Professions Education Unit, Hull York Medical School, York, UK
[3]Centre for Education and Innovation, Barts and The London School of Medicine and Dentistry, London, UK
[4]Health Sciences, University of York, York, UK
[5]Centre for Reviews and Dissemination, University of York, York, UK
[6]Department of Paediatric Oncology, Leeds Teaching Hospitals NHS Trust, Leeds, UK
[7]British Medical Association, London, UK

**Acknowledgements** The authors thank all the clinical academics who participated in this study. The authors also thank the wider research team involved in this large-scale research project for their contributions; the project steering group at the Clinical Academic Training Forum; and 'Women in Academic Medicine', and the British Medical Association for assistance in recruiting participants.

**Contributors** GF, PC, PT, MK, JM and JB conceived the study. GF, PC, PT, JB and JM obtained funding. ET, GF, JB, PC, ABK and AK collected data and conducted the analyses. GF, AK, JB, PT, CS and PC recruited participants and drafted the manuscript. ABK and AK curated the data. All authors reviewed the final manuscript. GF is the study guarantor.

**Funding** This publication presents independent research funded by Cancer Research UK (Award Number: C71037/A29824), The Academy of Medical Sciences, Health Education England, The Medical Research Council, The National Institute for Health Research (NIHR) and The Wellcome Trust. The views expressed are those of the author(s) and not necessarily those of The Academy of Medical Sciences, Cancer Research UK, Health Education England, The Medical Research Council, The National Institute for Health Research (NIHR), The Wellcome Trust or the Department of Health and Social Care.

**Competing interests** None declared.

**Patient and public involvement** Patients and/or the public were involved in the design, or conduct, or reporting, or dissemination plans of this research. Refer to the Methods section for further details.

**Patient consent for publication** Not applicable.

**Ethics approval** This study involves human participants and was approved by Hull York Medical School Ethics Committee (Ref: 19 32). Participants gave informed consent to participate in the study before taking part.

**Provenance and peer review** Not commissioned; externally peer reviewed.

**Data availability statement** No data are available. Any such request may be subject to additional ethical approvals. Deidentified transcripts would be made securely available to other researchers if a written case for access is made to the research team and permission is granted by the funders of the study. This should be sent directly to the corresponding author GF.

**ORCID iDs**
Gabrielle M Finn http://orcid.org/0000-0002-0419-694X
Paul Crampton http://orcid.org/0000-0001-8744-930X
Abisola Olatokunbo Balogun http://orcid.org/0000-0002-8177-9081
Jessica Elizabeth Morgan http://orcid.org/0000-0001-8087-8638
Carmen Soto http://orcid.org/0000-0002-6323-3069

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
