## [Reviewer comments · BMJ Open]

ARTICLE DETAILS

TITLE (PROVISIONAL)	The impact of the COVID-19 pandemic on the research activity and working experience of clinical academics, with a focus on gender and ethnicity: a qualitative study in the UK
AUTHORS	Finn, Gabrielle; Crampton, Paul; Buchanan, John; Balogun, Abisola Olatokunbo; Tiffin, Paul; Morgan, Jessica; Taylor, Ellie; Soto, Carmen; Kehoe, Amelia

VERSION 1 – REVIEW

REVIEWER	Ozga, Dorota Rzeszow University, Institute of Health Sciences, College of Medical Sciences
REVIEW RETURNED	13-Oct-2021

GENERAL COMMENTS	Congratulations on the idea, please complete the checklist with the methodology and include it in the supplementary materials.
--

REVIEWER	Das, Payal ICMR, Epidemiology and Communicable Diseases
REVIEW RETURNED	23-Oct-2021

GENERAL COMMENTS	The author should increase the sample size in order to understand the effect of the pandemic on the research activities and others. Focus Group Discussion(s) should be included as one of the method to interact with the participants.
---

REVIEWER	Minichil, Woredaw University of Gondar College of Medicine and Health Sciences, Psychiatry
REVIEW RETURNED	08-Nov-2021

GENERAL COMMENTS	1. English language and grammar usage should be improved2. From the four major themes, nothing is said about “fears and uncertainty” in the abstract.3. Strength and limitations should be separated and written in the form of paragraph (avoid bullets)4. Please align the right and left margin of the text5. Introduction is very limited; it doesn't exhaustively address literatures.
---

	6. Do authors apply the two data collection methods i.e., audio and written diary and interview, in a single participant? Or if parts of participants used one of them, how can they make sure its consistency? 7. Conclusion is not clear
--	--

REVIEWER	Ranieri, Veronica Tavistock and Portman NHS Foundation Trust, Research & Development
REVIEW RETURNED	18-Dec-2021

GENERAL COMMENTS	Thank you for the opportunity to review your paper. The paper was well-written and provided timely information regarding clinical academics' experiences during the initial waves of COVID in the UK. I suggest that the paper be accepted subject to some changes. The title is quite strong and doesn't feel like it quite matches the results which, aside from theme 3, read as a more generalised experience of all clinical academics – not of just women. Please could you revise the title? Was saturation reached? The authors spoke of enlarging the sample but it is not clear whether saturation was taken into account. Were there differences in the richness of responses between audio and written diary entries, and interviews? Did participants' accounts differ according to data collection method? 82 individuals participated, 30 provided audio, 68 interviewed... I couldn't quite tally these figures up. Could the authors state who provided X only, who provided both etc or perhaps add it to the table? How did the authors gain informed consent? The study title asks where all the women are, however, it's not always clear from the themes as to whether these are experienced more by people who identify as women or men (or other). Were there some themes that women or men identified more with outside of theme 3? Was free time (in new and emerging opportunities theme) a frequent occurrence? Was it gender-specific? Did participants' responses differ according to discipline, stage of training, age or other factor? Could the authors provide some headings to the discussion (ie. summary of findings, implications etc)? Could they address the strengths and limitations of the study? I'm struck by the quote on p.15 that starts with "the pro-vice chancellor had sent an email..." It sounds like this speaks to some pre-existing clinical academic or societal tensions. It feels like a
--

	missed opportunity to not discuss the implications of male fragility and how women’s continued fight for equality within the workplace is likely to lead to discomfort from those who have typically held most privilege, within the discussion. The authors alluded to there being misconceptions. What is meant by misconception? Were there some that caused greater difficulty in particular? Was there an indication of how clinical academics felt about potentially returning to their academic duties in future? Typo – p.5 Methods, Ethics, shows as ‘XXX’ Medical School Ethics. I initially assumed this was for blinding but perhaps not as authors’ names are listed? Could the authors please reword the final sentence of the data analysis starting with “In order to....” I think there might be a word missing? Typo p. 18, Returning to the frontline quote, Sentence 6, “but a lot of isolation and shielding”?
--	---

VERSION 1 – AUTHOR RESPONSE

Reviewer: 1

Dr. Dorota Ozga, Rzeszow University

Comments to the Author:

Dear Authors,

Congratulations on the idea, please complete the checklist with the methodology and include it in the supplementary materials.

Thank you – the COREQ checklist is included.

Reviewer: 2

Dr. Payal Das, ICMR

Comments to the Author:

The author should increase the sample size in order to understand the effect of the pandemic on the research activities and others.

Qualitative research does not rely on power. We have data from a carefully selected purposive sample, and the number of data points from multiple diary entries is significant. It would not be appropriate to collect more data.

Focus Group Discussion(s) should be included as one of the method to interact with the participants.

This was an audio diary and interview study. Focus groups were not appropriate for discussing instances of sexual, gender and racial discrimination.

Reviewer: 3

Dr. Woredaw Minichil, University of Gondar College of Medicine and Health Sciences

Comments to the Author:

1. English language and grammar usage should be improved.

Reviewed

2. From the four major themes, nothing is said about “fears and uncertainty” in the abstract.

Added

3. Strength and limitations should be separated and written in the form of paragraph

(avoid bullets) [NOTE FROM THE EDITORS: this reviewer suggestion is not correct, please follow journal formatting directions]

We note the editors comment and have not edited the text

4. Please align the right and left margin of the text

Journal formatting followed

5. Introduction is very limited; it doesn't exhaustively address literatures.

We kept the introduction shorter in order to maximise word count for results and discussion.

6. Do authors apply the two data collection methods I.e., audio and written diary and interview, in a single participant? Or if parts of participants used one of them, how can they make sure its consistency?

As noted within the paper - Eighty-two individuals participated, 30 provided audio diary data, 68 were interviewed, while 16 provided both.

7. Conclusion is not clear.

Revised

Reviewer: 4

Dr. Veronica Ranieri, Tavistock and Portman NHS Foundation Trust, University College London

Comments to the Author:

Thank you for the opportunity to review your paper. The paper was well-written and provided timely information regarding clinical academics' experiences during the initial waves of COVID in the UK. I suggest that the paper be accepted subject to some changes.

Thank you

The title is quite strong and doesn't feel like it quite matches the results which, aside from theme 3, read as a more generalised experience of all clinical academics – not of just women. Please could you revise the title?

Title edited - there was a differential impact for women which was notably more negative.

Was saturation reached? The authors spoke of enlarging the sample but it is not clear whether saturation was taken into account.

In the study we collected a very large amount of qualitative data. Saturation is no longer used as a marker or endpoint in qualitative research

(https://onlinelibrary.wiley.com/doi/epdf/10.1111/medu.13124?saml_referrer)

Instead, we collected data until samples were deemed to be adequate in terms of being of sufficient size to allow transfer-ability to other contexts), appropriate (i.e. in terms of data being able to answer research questions), and aligned with their research questions and methodological orientation. Malterud ref on information power:

Malterud, K., Siersma, V.D. and Guassora, A.D., 2016. Sample size in qualitative interview studies: guided by information power. *Qualitative health research*, 26(13), pp.1753-1760.

Were there differences in the richness of responses between audio and written diary entries, and interviews? Did participants' accounts differ according to data collection method?

All data is rich and useful. A line has been added to the text.

Audio diaries varied in length and thus depth, however, there was no variability in the richness of data.

82 individuals participated, 30 provided audio, 68 interviewed... I couldn't quite tally these figures up. Could the authors state who provided X only, who provided both etc or perhaps add it to the table?

We discussed this in the written text, 16 provided both. It is not possible to pair the data and demographics and further than what is provided in the text and table for those who conducted both as it may lead to the individuals being identifiable.

How did the authors gain informed consent?

This was in the body of the paper – we inserted 'electronic' for clarity

The study title asks where all the women are, however, it's not always clear from the themes as to whether these are experienced more by people who identify as women or men (or other). Were there some themes that women or men identified more with outside of theme 3?

The findings reveal the range of experiences identified from the participants. Where relevant we have highlighted the relevant themes in order to make the links clearer to our conclusions and title. The title has been edited too.

Was free time (in new and emerging opportunities theme) a frequent occurrence? Was it gender-specific?

Additional text added - Both men and women reported having some additional free-time, although this was contradictory to comments with respect to childcare and the burden of extra work.

Did participants' responses differ according to discipline, stage of training, age or other factor?

We collected a very large amount of data for a qualitative project yet this would not allow us to make causal links between factors and experiences. The value of the study is through revealing the factors involved and their potential impacts.

Could the authors provide some headings to the discussion (ie. summary of findings, implications etc)? Could they address the strengths and limitations of the study?

The article has a strengths and limitations section already. It follows the abstract as per the author guidelines.

I'm struck by the quote on p.15 that starts with "the pro-vice chancellor had sent an email..." It sounds like this speaks to some pre-existing clinical academic or societal tensions. It feels like a missed opportunity to not discuss the implications of male fragility and how women's continued fight for equality within the workplace is likely to lead to discomfort from those who have typically held most privilege, within the discussion.

Male fragility has been briefly included in the discussion but we would not want to over interpret this.

The authors alluded to there being misconceptions. What is meant by misconception? Were there some that caused greater difficulty in particular?

Text edited – please note the misconceptions are also listed in the table. The quotes under the theme also exemplify the misconceptions.

Was there an indication of how clinical academics felt about potentially returning to their academic duties in future?

This was not something that was identified as a theme within in our data.

Typo – p.5 Methods, Ethics, shows as 'XXX' Medical School Ethics. I initially assumed this was for blinding but perhaps not as authors' names are listed?

This was for review, which we believed to have been blinded. Name inserted.

Could the authors please reword the final sentence of the data analysis starting with "In order to..." I think there might be a word missing?

No word missing but edited for clarity.

Typo p. 18, Returning to the frontline quote, Sentence 6, "but a lot of isolation and shielding"?

If changed to of

VERSION 2 – REVIEW

REVIEWER	Ranieri, Veronica Tavistock and Portman NHS Foundation Trust, Research & Development
REVIEW RETURNED	15-Feb-2022

GENERAL COMMENTS	*"Was saturation reached? The authors spoke of enlarging the sample but it is not clear whether saturation was taken into account." "In the study we collected a very large amount of qualitative data. Saturation is no longer used as a marker or endpoint in qualitative research (https://onlinelibrary.wiley.com/doi/epdf/10.1111/medu.13124?saml_referrer). Instead, we collected data until samples were deemed to be adequate in terms of being of sufficient size to allow transfer-ability to other contexts), appropriate (i.e. in terms of data being able to answer research questions), and aligned with their research questions and methodological orientation. Malterud ref on information power: Malterud, K., Siersma, V.D. and Guassora, A.D., 2016. Sample size in qualitative interview studies: guided by information power. Qualitative health research, 26(13), pp.1753-1760. " - There is ongoing debate within the literature regarding this. I would have appreciated some reflection from the authors regarding whether they felt that there were some sub-groups/themes which felt more saturated than others. *"Did participants' responses differ according to discipline, stage of training, age or other factor?" "We collected a very large amount of data for a qualitative project yet this would not allow us to make causal links between factors and experiences. The value of the study is through revealing the factors involved and their potential impacts." - Again, it would be helpful if the authors provided some reflection on this as experiences may differ between more junior and senior clinical academics and between specialties (ie. is it possible that an anaesthetics clinical academic may have had a different experience to a psychiatry or dermatology clinical academic during covid peaks, for instance in terms of deployment or work stresses?). *"Could the authors provide some headings to the discussion (ie. summary of findings, implications etc)? Could they address the strengths and limitations of the study?" "The article has a strengths and limitations section already. It follows the abstract as per the author guidelines." - The lack of headings makes the discussion difficult to read. Please could this be amended? *"I'm struck by the quote on p.15 that starts with "the pro-vice chancellor had sent an email..." It sounds like this speaks to some pre-existing clinical academic or societal tensions. It feels like a missed opportunity to not discuss the implications of male fragility and how women's continued fight for equality within the workplace is likely to lead to discomfort from those who have typically held most privilege, within the discussion." "Male fragility has been briefly included in the discussion but we would not want to over interpret this." - I invite the authors to reflect on this using their own words rather than mine - or in disagreeing should they feel that way.
--

VERSION 2 – AUTHOR RESPONSE

Reviewer: 4
 Dr. Veronica Ranieri, Tavistock and Portman NHS Foundation Trust, University College London

Comments to the Author:

*"Was saturation reached? The authors spoke of enlarging the sample but it is not clear whether saturation was taken into account."

"In the study we collected a very large amount of qualitative data. Saturation is no longer used as a marker or endpoint in qualitative research

(https://onlinelibrary.wiley.com/doi/epdf/10.1111/medu.13124?saml_referrer). Instead, we collected data until samples were deemed to be adequate in terms of being of sufficient size to allow transferability to other contexts), appropriate (i.e. in terms of data being able to answer research questions), and aligned with their research questions and methodological orientation. Malterud ref on information power: Malterud, K., Siersma, V.D. and Guassora, A.D., 2016. Sample size in qualitative interview studies: guided by information power. *Qualitative health research*, 26(13), pp.1753-1760. "

- There is ongoing debate within the literature regarding this. I would have appreciated some reflection from the authors regarding whether they felt that there were some sub-groups/themes which felt more saturated than others.

Response: As we outlined in our previous response, saturation is no longer a consideration within qualitative research. Braun and Clark have also written about this recently: <https://www.tandfonline.com/doi/full/10.1080/2159676X.2019.1704846>

"...But when it comes to reflexive TA, data saturation is not a particularly useful, or indeed theoretically coherent, concept.5 Other concepts – like information power – can offer a more useful way of thinking through data samples. But we recognise that data saturation might be a concept reflexive TA researchers pragmatically chose to deploy to appease research gatekeepers, or might be required to. In doing so, they (and indeed we) are, however, complicit in perpetuating the myth of data saturation as a vital rationale and practice for qualitative research more generally. If a claim of data saturation must be deployed for reflexive TA to ‘pass go’, we encourage researchers to critically comment on this, or provide some justification for it. Or, indeed, perhaps to re-theorise (data) saturation in new, exciting, and currently unanticipated ways"

As one doesn't seek to quantify themes in qualitative research, describing some themes as more saturated would be meaningless. We have added text to the manuscript to explain our stance on this.

*"Did participants' responses differ according to discipline, stage of training, age or other factor?"

"We collected a very large amount of data for a qualitative project yet this would not allow us to make causal links between factors and experiences. The value of the study is through revealing the factors involved and their potential impacts."

- Again, it would be helpful if the authors provided some reflection on this as experiences may differ between more junior and senior clinical academics and between specialties (ie. is it possible that an anaesthetics clinical academic may have had a different experience to a psychiatry or dermatology clinical academic during covid peaks, for instance in terms of deployment or work stresses?).

Response: This was not the aim or a research question in this study, nor did participants discuss their specialities in the context of covid enough for us to meaningfully interpret such data. Had this have been the case, it would have been included.

*"Could the authors provide some headings to the discussion (ie. summary of findings, implications etc)? Could they address the strengths and limitations of the study?"

"The article has a strengths and limitations section already. It follows the abstract as per the author guidelines."

- The lack of headings makes the discussion difficult to read. Please could this be amended?

Response: Sub-headings added

*"I'm struck by the quote on p.15 that starts with "the pro-vice chancellor had sent an email..." It sounds like this speaks to some pre-existing clinical academic or societal tensions. It feels like a missed opportunity to not discuss the implications of male fragility and how women's continued fight for equality within the workplace is likely to lead to discomfort from those who have typically held most privilege, within the discussion."

"Male fragility has been briefly included in the discussion but we would not want to over interpret this."-

I invite the authors to reflect on this using their own words rather than mine - or in disagreeing should they feel that way.

Response: Edited - we have also gone back through the data and added more quotes to this effect. It is framed more in terms of positive discrimination but we can see the argument about male fragility and agree with your framing.

(Previous submission:Response to round 1 of reviewer comments)

Thank you for the opportunity to revise our paper. We have completed all revisions as requested.

Editor revisions:

*Please spell out abbreviations in the abstract (eg, CA)

Done

*In the Abstract, please state how many provided audio diaries and how many were interviewed, and in view of the conclusion about women and ethnicity, the numbers (and %) of men and women, and also ethnicity data.

Done

*Please explain how the sample was selected? Was it exhaustive? Purposive?

All within the COREQ

Added to abstract and main body of text

*The conclusion in the Abstract needs to be more balanced about the four themes of the research.

Edited

*In the main body of the text, please use less emotive language (like 'alarming').

Edited

*Details of ethics approval are missing (name of ethics committee, reference number for approval, etc).

This was removed for blinded review – now reinstated

*In the Introduction, can the authors be clear why a qualitative approach was selected? Can they be clear in the objective whether or not ethnicity and women's issues were specifically addressed?

*Please provide more information about recruitment. Was there purposive sampling?

Added

*Please provide more detail on the data analysis, including how many authors reviewed the data.

All within COREQ and text edited

Reviewer: 1

Dr. Dorota Ozga, Rzeszow University

Comments to the Author:

Dear Authors,

Congratulations on the idea, please complete the checklist with the methodology and include it in the supplementary materials.

Thank you – the COREQ checklist is included.

Reviewer: 2

Dr. Payal Das, ICMR

Comments to the Author:

The author should increase the sample size in order to understand the effect of the pandemic on the research activities and others.

Qualitative research does not rely on power. We have data from a carefully selected purposive sample, and the number of data points from multiple diary entries is significant. It would not be appropriate to collect more data.

Focus Group Discussion(s) should be included as one of the method to interact with the participants. This was an audio diary and interview study. Focus groups were not appropriate for discussing instances of sexual, gender and racial discrimination.

Reviewer: 3

Dr. Woredaw Minichil, University of Gondar College of Medicine and Health Sciences

Comments to the Author:

1. English language and grammar usage should be improved.

Reviewed

2. From the four major themes, nothing is said about “fears and uncertainty” in the abstract.

Added

3. Strength and limitations should be separated and written in the form of paragraph (avoid bullets) [NOTE FROM THE EDITORS: this reviewer suggestion is not correct, please follow journal formatting directions]

We note the editors comment and have not edited the text

4. Please align the right and left margin of the text

Journal formatting followed

5. Introduction is very limited; it doesn't exhaustively address literatures.

We kept the introduction shorter in order to maximise word count for results and discussion.

6. Do authors apply the two data collection methods i.e., audio and written diary and interview, in a single participant? Or if parts of participants used one of them, how can they make sure its consistency?

As noted within the paper - Eighty-two individuals participated, 30 provided audio diary data, 68 were interviewed, while 16 provided both.

7. Conclusion is not clear.

Revised

Reviewer: 4

Dr. Veronica Ranieri, Tavistock and Portman NHS Foundation Trust, University College London

Comments to the Author:

Thank you for the opportunity to review your paper. The paper was well-written and provided timely information regarding clinical academics' experiences during the initial waves of COVID in the UK. I suggest that the paper be accepted subject to some changes.

Thank you

The title is quite strong and doesn't feel like it quite matches the results which, aside from theme 3, read as a more generalised experience of all clinical academics – not of just women. Please could you revise the title?

Title edited - there was a differential impact for women which was notably more negative.

Was saturation reached? The authors spoke of enlarging the sample but it is not clear whether saturation was taken into account.

In the study we collected a very large amount of qualitative data. Saturation is no longer used as a marker or endpoint in qualitative research

(https://onlinelibrary.wiley.com/doi/epdf/10.1111/medu.13124?saml_referrer)

Instead, we collected data until samples were deemed to be adequate in terms of being of sufficient size to allow transfer-ability to other contexts), appropriate (i.e. in terms of data being able to answer

research questions), and aligned with their research questions and methodological orientation. Malterud ref on information power:
Malterud, K., Siersma, V.D. and Guassora, A.D., 2016. Sample size in qualitative interview studies: guided by information power. *Qualitative health research*, 26(13), pp.1753-1760.

Were there differences in the richness of responses between audio and written diary entries, and interviews? Did participants' accounts differ according to data collection method?

All data is rich and useful. A line has been added to the text.

Audio diaries varied in length and thus depth, however, there was no variability in the richness of data.

82 individuals participated, 30 provided audio, 68 interviewed... I couldn't quite tally these figures up. Could the authors state who provided X only, who provided both etc or perhaps add it to the table?

We discussed this in the written text, 16 provided both. It is not possible to pair the data and demographics and further than what is provided in the text and table for those who conducted both as it may lead to the individuals being identifiable.

How did the authors gain informed consent?

This was in the body of the paper – we inserted 'electronic' for clarity

The study title asks where all the women are, however, it's not always clear from the themes as to whether these are experienced more by people who identify as women or men (or other). Were there some themes that women or men identified more with outside of theme 3?

The findings reveal the range of experiences identified from the participants. Where relevant we have highlighted the relevant themes in order to make the links clearer to our conclusions and title.

The title has been edited too.

Was free time (in new and emerging opportunities theme) a frequent occurrence? Was it gender-specific?

Additional text added - Both men and women reported having some additional free-time, although this was contradictory to comments with respect to childcare and the burden of extra work.

Did participants' responses differ according to discipline, stage of training, age or other factor?

We collected a very large amount of data for a qualitative project yet this would not allow us to make causal links between factors and experiences. The value of the study is through revealing the factors involved and their potential impacts.

Could the authors provide some headings to the discussion (ie. summary of findings, implications etc)? Could they address the strengths and limitations of the study?

The article has a strengths and limitations section already. It follows the abstract as per the author guidelines.

I'm struck by the quote on p.15 that starts with "the pro-vice chancellor had sent an email..." It sounds like this speaks to some pre-existing clinical academic or societal tensions. It feels like a missed opportunity to not discuss the implications of male fragility and how women's continued fight for equality within the workplace is likely to lead to discomfort from those who have typically held most privilege, within the discussion.

Male fragility has been briefly included in the discussion but we would not want to over interpret this.

The authors alluded to there being misconceptions. What is meant by misconception? Were there some that caused greater difficulty in particular?

Text edited – please note the misconceptions are also listed in the table. The quotes under the theme also exemplify the misconceptions.

Was there an indication of how clinical academics felt about potentially returning to their academic duties in future?

This was not something that was identified as a theme within in our data.

Typo – p.5 Methods, Ethics, shows as ‘XXX’ Medical School Ethics. I initially assumed this was for blinding but perhaps not as authors’ names are listed?

This was for review, which we believed to have been blinded. Name inserted.

Could the authors please reword the final sentence of the data analysis starting with “In order to...” I think there might be a word missing?

No word missing but edited for clarity.

Typo p. 18, Returning to the frontline quote, Sentence 6, “but a lot of isolation and shielding”?

If changed to of

VERSION 3 – REVIEW

REVIEWER	Ranieri, Veronica Tavistock and Portman NHS Foundation Trust, Research & Development
REVIEW RETURNED	20-Mar-2022
GENERAL COMMENTS	Thank you for your addressed changes.